# Ringed seal (*Pusa hispida*) breeding habitat on the landfast ice in northwest Alaska during spring 1983 and 1984

**Donna D. W. Hauser**[1]*, **Kathryn J. Frost**[2], **John J. Burns**[3]

**1** International Arctic Research Center, University of Alaska Fairbanks, Fairbanks, Alaska, United States of America, **2** Alaska Department of Fish and Game (retired), Kailua Kona, Hawaii, United States of America, **3** Living Resources, Inc., Fairbanks, Alaska, United States of America

* dhauser2@alaska.edu

## Abstract

There has been significant sea ice loss associated with climate change in the Pacific Arctic, with unquantified impacts to the habitat of ice-obligate marine mammals such as ringed seals (*Pusa hispida*). Ringed seals maintain breathing holes and excavate subnivean lairs on sea ice to provide protection from weather and predators during birthing, nursing, and resting. However, there is limited baseline information on the snow and ice habitat, distribution, density, and configuration of ringed seal structures (breathing holes, simple haul-out lairs, and pup lairs) in Alaska. Here, we describe historic field records from two regions of the eastern Chukchi Sea (Kotzebue Sound and Ledyard Bay) collected during spring 1983 and 1984 to quantify baseline ringed seal breeding habitat and map the distribution of ringed seal structures using modern geospatial tools. Of 490 structures located on pre-established study grids by trained dogs, 29% were pup lairs (25% in Kotzebue Sound and 33% in Ledyard Bay). Grids in Ledyard Bay had greater overall density of seal structures than those in Kotzebue Sound (8.6 structures/km² and 7.1 structures/km²), but structures were larger in Kotzebue Sound. Pup lairs were located in closer proximity to other structures and characterized by deeper snow and greater ice deformation than haul-out lairs or simple breathing holes. At pup lairs, snow depths averaged 74.9 cm (range 37–132 cm), with ice relief nearby averaging 76 cm (range 31–183 cm), and ice deformation 29.9% (range 5–80%). We compare our results to similar studies conducted in other geographic regions and discuss our findings in the context of recent declines in extent and duration of seasonal cover of landfast sea ice and snow deposition on sea ice. Ultimately, additional research is needed to understand the effects of recent environmental changes on ringed seals, but our study establishes a baseline upon which future research can measure pup habitat in northwest Alaska.

## Introduction

Climate change in the Pacific Arctic, particularly the Chukchi and Beaufort Seas, has been marked by unprecedented reductions in sea ice extent, thickness, and duration of seasonal

**Data Availability Statement:** Data are available in a DataOne repository here: https://search.dataone.org/view/10.24431/rw1k5b3. Hauser, D.D.W. and K.J. Frost. 2021. Ringed seal structure measurements, types, snow and ice habitat, and

evidence of predation collected during spring 1983 and 1984. DOI: 10.24431/rw1k5b3.

**Funding:** Analysis of these data was funded by the North Pacific Research Board, Grant No. 1811, to D.D.W.H. and K.J.F. Data were originally collected with funding from the Outer Continental Shelf Environmental Assessment Program, contract NA-81-RAC-00045 RU#232 to J.J.B. and K.J.F. The funders had no role in study design, data collection and analysis, decision to publish, or preparation of the manuscript.

**Competing interests:** The authors have declared no competing interests exist.

cover [1–3]. Sea ice loss is expected to directly affect ice-dependent Arctic marine mammals [4], although complicated responses among species and populations are increasingly recognized in the Pacific Arctic region [e.g. 5–7]. For ice-obligate species, such as ringed seals (*Pusa hispida*), sea ice is essential habitat as a platform for reproduction, molting, and resting [8,9].

Ringed seals are a small, numerous, and circumpolar phocid that is found across the Arctic [10,11]. As key components of Arctic marine ecosystems, ringed seals are generalist consumers as well as primary prey of polar bears (*Ursus maritimus*) and Arctic foxes (*Vulpes lagopus*) [12,13]. Ringed seals are also a traditional resource for coastal Inuit people who continue to rely on ringed seals and other marine mammals for nutritional, cultural and spiritual purposes [14]. Ringed seals thus reflect ecosystem health as sentinels of environmental change [15,16], ultimately playing important roles in shifting ecological interactions and food web productivity in response to the loss of sea ice [17–19]. Understanding how changing snow and ice regimes are impacting these ice-obligate seals is relevant to the health of both Arctic marine ecosystems and coastal communities.

Adult ringed seals over-winter in areas of landfast and dense pack ice where they maintain breathing holes and excavate lairs in snow on top of the ice [11]. Subadults, unconstrained by the need to use breeding habitat in stable landfast ice, use pack ice and areas near the ice edge during winter and spring [20]. Sea ice with substantial snow cover is necessary for construction of subnivean lairs, so the structure of both ice and snow are important habitat features [8]. In particular, wind-drifted snow accumulates in the lee of pressure ridges and other deformations of the ice. Lairs are classified as simple haul-out lairs, used for resting by adult male or female seals, and pup lairs in which adult females give birth in April and May, followed by 5–6 weeks of nursing [10,21]. Until snow melts and lairs collapse, the lairs provide protection from weather and predation [13,22–24].

More information is needed to understand the impacts of changing snow and ice conditions on ringed seals. In several Arctic regions, changes in sea ice extent, quality, and phenology or other environmental conditions have been linked to decreases in ringed seal body condition, pregnancy rates, and pup survival in addition to increases in stress hormones [e.g. 25–28], although impacts vary among populations and sub-regions [7,29]. Snow depth and the extent and thickness of sea ice have decreased in recent decades, accompanied by earlier spring melt [e.g. 30,31]. Modeling projections suggest that within this century snow depth may be insufficient for construction of ringed seal lairs in some parts of their range [32]. This could impact pup survival [9,33,34]. Inability to dig lairs or the premature collapse of lairs with thin roofs may result in pups born on top of the ice or early opening of lairs, likely decreasing survival by exposing the pups to increased predation or severe weather. Alternatively, seals may seek areas with more optimal snow and ice conditions, despite a predilection for breeding site fidelity, territoriality, and limited spring and winter movements from lairs [35–37]. Aerial surveys conducted in 1985–1987 and 1999–2000 indicated substantial annual variation in densities of ringed seals within the same sectors along the Chukchi Sea coastline, suggesting that suitability of snow and ice conditions moderates site selection in this highly variable environment [38,39].

There are few published baseline data on the snow and ice attributes of ringed seal lairs and associated breeding habitat [8,34,40], and none for Alaska, that can provide historical perspectives for comparison to modern conditions. There are also no fine-scale remotely-sensed snow on sea ice data products from historic periods. However, datasets exist from past studies that can be analyzed to document historical breeding structures and conditions. Here, we address questions about past conditions of ringed seal spring breeding habitat in northwest Alaska by analyzing field data collected nearly 40 years ago but not previously published. We describe fine-scale data on snow and ice associated with nearly 500 ringed seal structures (i.e. breathing

holes, simple haul-out lairs, and pup lairs). Data were collected at two locations on the landfast ice in the eastern Chukchi Sea during spring 1983 and 1984, prior to substantial sea ice loss that has occurred in recent decades. Our goals are to: 1) quantify ice and snow conditions at pup lairs and other ringed seal structures; (2) describe the sizes and regional composition of ringed seal structures; (3) estimate the density of structures in the two study regions; and 4) map the distribution and describe the spatial relationship of structures. These historic data can be used to establish a baseline for comparison with current and future ice and snow conditions and to better understand the impacts of decreasing snow and ice on ringed seal breeding habitat.

## Materials and methods

### Study area

Our studies were conducted in two coastal areas of the eastern Chukchi Sea. Searches for ringed seal structures were on landfast sea ice in southeastern Kotzebue Sound (~66˚16'N, 162˚32'W) in 1983 and ~170 nautical miles farther north in Ledyard Bay (~68˚53'N, 165˚50'W) east of Cape Lisburne in 1984 (Fig 1). The Chukchi Sea is a shallow (mean depth ~58 m) and productive continental shelf ecosystem connecting the sub-Arctic Bering Sea and Arctic Ocean. Kotzebue Sound is a large estuarine embayment (>90,000 km$^2$), located north of the Bering Strait in the southeast Chukchi Sea. The shallow waters (mean depth ~14 m) are influenced by four major rivers, including the Noatak, Kobuk, Selawik, and Buckland rivers. Ledyard Bay is a wide northward-facing bay located along the open northwest Alaska coast that is typical of coastal marine systems in the eastern Chukchi Sea. Historically, Ledyard Bay was covered by landfast ice during the ice-covered season with an active offshore lead system. During both years of our study, landfast ice started forming in early to mid-October in Kotzebue Sound and mid to late October in Ledyard Bay [41]. Following freeze-up, landfast ice thickens and extends seaward throughout autumn and winter. Historically, Kotzebue Sound and other large embayments became completely covered with landfast ice that was 1–2 m thick. Landfast ice in the Ledyard Bay region extended several tens of kilometers from shore.

A flaw zone develops at the interface of the stable landfast ice and the dynamic, drifting pack ice that forms on the ocean. A series of large pressure ridges often develops approximately parallel to this interface (called the shear zone). Some pressure ridges become grounded onto the seafloor, which stabilizes the landfast ice and generally limits deformation of the landfast ice closer to shore. Deformation of the landfast ice generally increases seaward. Wind-drifted snow accumulates in the lee of pressure ridges and other deformations of the ice.

### Detecting ringed seal structures

Camps were established on the ice or on land nearby. Our searches for seal structures were conducted in Kotzebue Sound during 5–29 April 1983 and in Ledyard Bay during 6 April to 13 May 1984 (Table 1). Two specially-trained Labrador retriever dogs were used to search for and indicate the presence of ringed seal structures, following procedures described in Smith and Stirling [8]. The dogs worked either singly or in pairs, ahead of a slow moving snowmachine along pre-established survey lines (as described below). When scent was detected, a dog moved upwind to its source and indicated the location by digging in the snow, usually close to the access hole (Fig 2). A wooden handled aluminum rod, ~1 cm in diameter and marked every 10 cm to a length of 150 cm, was used to probe the snow until the seal structure was located, as indicated by a void under the snow or a thin ice layer at a lair's surface. Structures were then opened, measured, and classified (as described below). All structures that were opened were subsequently reconstructed.

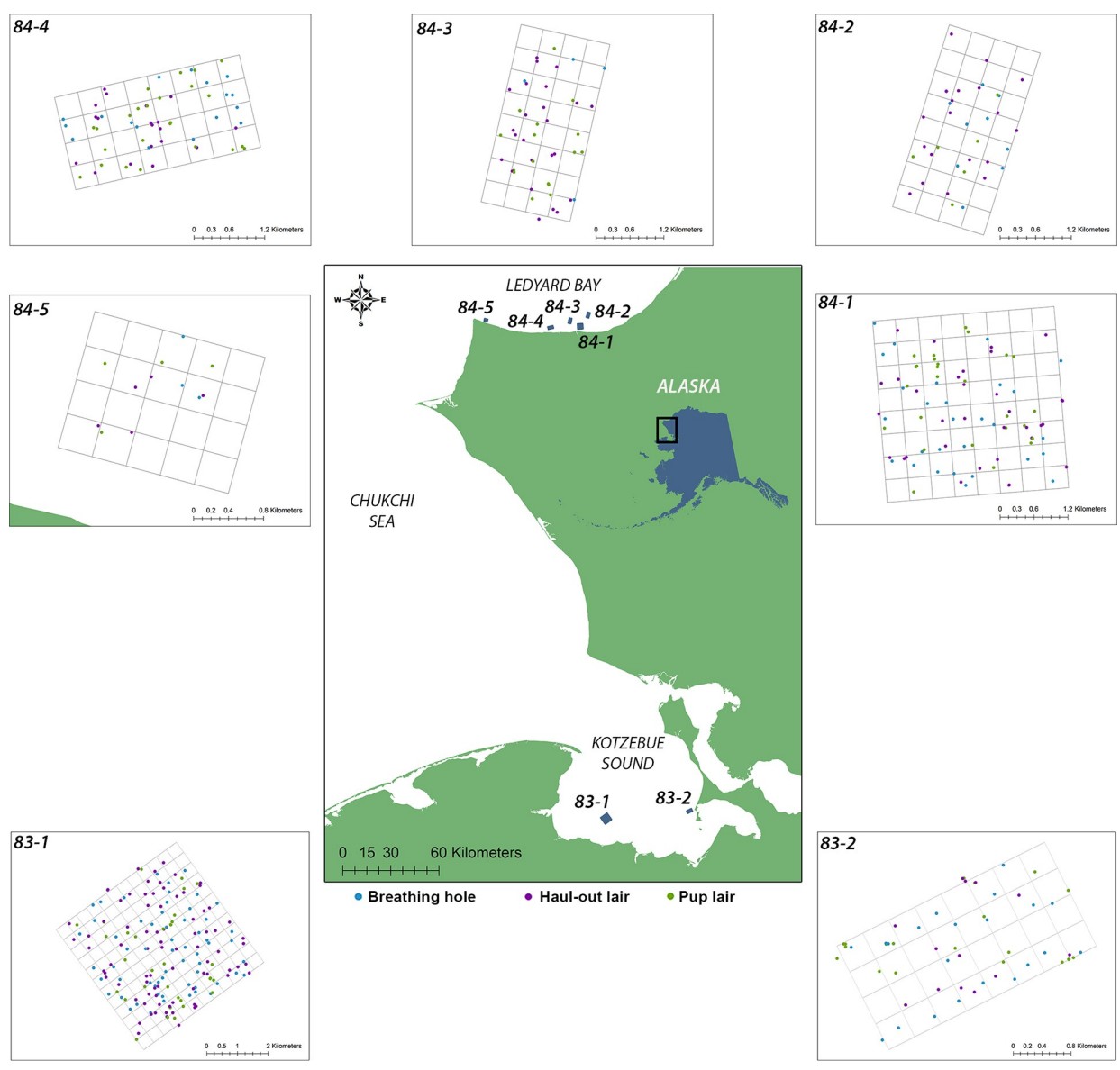

**Fig 1. Study areas in Kotzebue Sound (1983) and Ledyard Bay (1984).** Maps of survey grids include all ringed seal structures located by dogs.

Two survey grids (~33 km² total) were established in Kotzebue Sound and five (~31 km² total) in Ledyard Bay (Fig 1). The grids were intentionally located to sample ice habitats at different distances from shore and in different ice conditions. Field work occurred before the advent of handheld GPS units, so the location of each seal structure was recorded as a combination of paced distance and magnetic heading from the pre-established survey lines. Survey grids were composed of square-shaped 'blocks' that were based on parallel search lines spaced 440 yards (~402 m) apart as well as perpendicular cross lines, also 440 yards apart. Most grids contained at least one base line that generally traced an extensive feature of flat ice such as a frozen lead or flat pan. Each grid had at least one known 'anchor' point with a latitude and longitude, which was subsequently important for mapping the grids. Grid lines were established using a surveyor's transit, and the length of each line was measured using the snowmachine odometer. Lines were also checked against the Global Navigation System of the helicopter

**Table 1. Descriptive characteristics for grids surveyed for ringed seal structures in the Kotzebue Sound and Ledyard Bay regions of the Chukchi Sea in spring 1983 and 1984, respectively.**

| Region (Coordinates of grid centerpoint) | Grid | Dates surveyed | Distance from shore (km) | Comments and notable ice features |
|---|---|---|---|---|
| **Kotzebue Sound** | | | | |
| 66˚12'N 163˚09'W | 83–1 | 5–22 April 1983 | 14.8 | Variable ice conditions. Bisected E-W by refrozen lead 90–275 m wide. Perpendicular lead through west end of grid. Water depth 14 m and ice thickness 102 cm at center grid. Diverse ice conditions including a few extensive flat ice areas mixed among large areas of small flat pans interspersed among large fields of low rubble and some larger pressure ridges and jumbled ice. Overall 35% deformation and 60–90 cm relief with some areas of relief up to 200 cm and large deep snow drifts. |
| 66˚19'N 162˚01'W | 83–2 | 25–30 April 1983 | 3.7 | Melt process was well advanced during survey. Water depth 11–13 m, ice thickness 120–130 cm. Extensive areas of flat ice surrounded by irregular and low-relief pressure ridges. SW edge bordered by flat, undeformed ice with very little snow. Minimal snow cover overall, except for drifts near pressure ridges. Overall 10–20% deformation and average 30–60 cm relief, but melt advanced and these values may not reflect earlier conditions. |
| **Ledyard Bay** | | | | |
| 68˚57'N 164˚34'W | 84–1 | 6–19 April 1984 | 3.0 | Fairly uniform ice conditions other than one moderate (up to 4 m high) pressure ridge. Other relief to 1.5 m, most 60–90 cm. Deformation generally 10–30%, up to 80%. |
| 69˚01'N 164˚29'W | 84–2 | 17–26 April 1984 | 10.4 | Seaward of a large pressure ridge. Jumbled ice and several pressure ridges 2.4–3 m. Otherwise relief to 2.1 m, most 0.6–1 m. Deformation generally 10–30%, up to 80%. |
| 68˚58'N 164˚45'W | 84–3 | 24 April-3 May 1984 | 7.4 | Adjacent to large area of flat ice and shoreward of a large pressure/jumble ridge. Relief to 2.4 m, most 0.6–1.8 m. Deformation mostly 10–30%, up to 80–100%. |
| 68˚54'N 165˚02'W | 84–4 | 30 April-7 May 1984 | 3.1 | Shoreward of large pressure/jumble ridge up to 6 m tall. Variable relief within the grid, up to 3 m but mostly 0.6–1.5 m with some small flat areas. Deformation variable, 10–20% mixed with areas of 40–60%. |
| 68˚53'N 166˚02'W | 84–5 | 8–12 May | 1.9 | Just offshore of flat shore ice, and inshore of a large pressure ridge. Relief to 2.1 m, mostly 0.3–1 m. Deformation mostly 10–30%, to 60%. Offshore from a dump and 2–4 km east of the Cape Lisburne Long Range Radar Station. Highly disturbed and dirty due to proximity to the dump, facility, and airfield. |

servicing the field sites as well as by running the snowmachine over courses of known length (e.g. an air field). In the field, linear measurements were recorded in statute miles and yards, and later converted to meters. Each search line was marked every 440 yards (~402 m) with colored and coded stakes, which were the fixed points from which the distance and bearing of each seal structure was measured.

Search efficiency of the two dogs was evaluated by comparing mean and maximum detection distances, types of structures found, and success under different wind conditions. The two dogs were similar in performance and showed no bias in the types of structures they located [41]. Despite the perception that wind affects a dog's ability to locate structures, the dogs were equally effective at moderate wind speeds and all angles between wind and search lines. The proportion of seal structures found during first-time line searches was ~65% of the total number eventually found, with a conservative effective survey strip width of 200 m on the windward side of a line [41]. Each grid line was searched at least twice, traversed in both directions to minimize the effect of wind direction, and most lines were searched repeatedly until no additional structures were found. Conditions of low wind speed and soft wet snow, blowing snow, or very rough ice all limited the dogs' ability to locate structures.

## Classification and measurements of seal structures

We classified four types of structures: 1) 'breathing holes' through the ice that were not used for hauling out, 2) 'haul-out' lairs that were single-chambered cavities excavated in the snow over an enlarged breathing hole, sometimes called 'simple' lairs, 3) 'complex' lairs that contained multiple chambers, and 4) 'pup' lairs that were multi-chambered cavities with positive

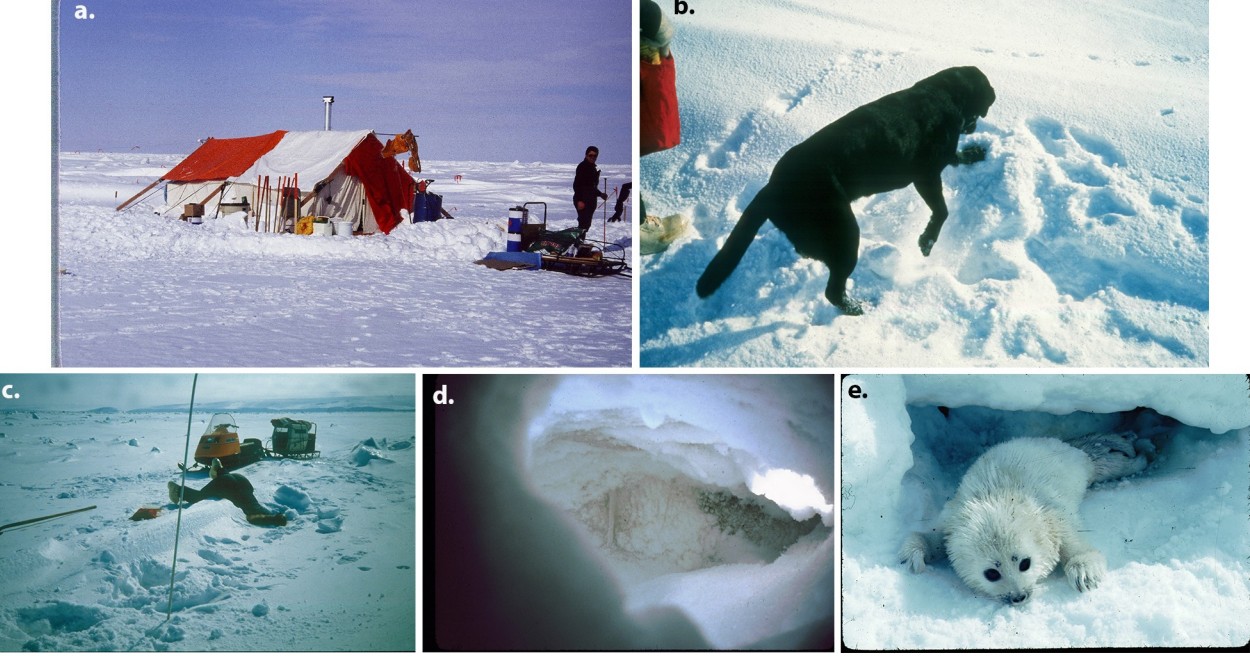

**Fig 2. Photos illustrating field activities during spring 1983 and 1984.** Photos include: (a) field camp associated with Kotzebue Sound grid 83–1. Stakes used to mark locations within a grid are shown along the side of the tent (photo by K.J.F.); (b) "Charlie", a Labrador retriever, indicates the location of a seal structure (photo by K.J.F.); (c) K.J.F. measuring dimensions of a structure in a linear ridge feature (photo by Sue Hills); (d) the inside of a seal lair (photo by Ann Adams); (e) ringed seal pup at a partially collapsed lair (photo by Lloyd Lowry). Photos reprinted with permission from Kathryn J. Frost.

evidence of a pup's presence, including an actual pup, remains of a dead pup, blood or afterbirth, pup fur referred to as lanugo, pup claw marks on walls, or tunnels and chambers excavated by a pup. Lairs all start as simple oval-shaped haul-out lairs created by an adult seal. Lairs used by females with pups are enlarged either before or once a pup is born. This process has not been observed. Some lairs contain 5–10 chambers that are presumably created as pups grow, become mobile, and dig tunnels within a lair. In the field, lairs were only classified as 'pup lairs' when a pup or direct evidence of a pup was observed. For analysis, we assumed complex lairs were also used for or by pups, so complex and pup lairs were combined for statistical analyses and are hereafter referred to as 'pup lairs.' In very few instances (n = 4), the type of subnivean structure could not be classified when it was located in a thick ice jumble or pressure ridge and could not be excavated.

Measurements taken at each structure included: diameter of breathing or access hole; maximum length, width, and height of lairs; snow depth, as measured to the ice surface near the center of the structure; average height of ice relief; and the extent of ice deformation (0–100%) within a 200 m radius. Ice relief and deformation are indicators of habitat complexity related to topography of the ice surface affecting snow accumulation. Higher values indicate the presence of pressure ridges that facilitate snow accumulation; lower values indicate flatter, less deformed ice, and potential wind scouring or less structure for snow deposition. Ice relief was generally estimated in 1-foot categories. If a range of ice relief was recorded (e.g. 1–2 ft), the midpoint was used and converted to metric in our analyses. The extent of ice deformation within a 200 m radius of the structure was visually scanned as the aerial percentage of ice that was deformed. Ice deformation was often estimated as a range (e.g. 10–20%) and the midpoint used in our analyses.

## Spatial analyses

Recent advances in geographic information systems (GIS) allow more accurate mapping and assessment of the spatial distribution and configuration of structures detected. We used Arc-GIS v. 10.6 (ESRI, Redlands, CA), applying an Alaska Albers equal-area conic projection, to map search grids as well as to estimate locations of the detected seal structures based on pacing and angles. Mapping each grid involved first creating a shapefile for the known anchor points, which were then used to create a regular grid using the 'Create Fishnet' tool (in the Data Management toolbox). Based on the hand-drawn maps and calculations for grid corner points, each grid was spatially-referenced to the known anchor point as the origin and rotated according to a calculated corner point. The number of rows and columns were specified to create cells that were 402.336 m on each side (corresponding to the 440 yard edge of each grid used in the field). Grid 83–1 in Kotzebue Sound was the only grid that required additional procedures, because the known anchor point for 'camp' was not a corner. Four separate sub-units were created for grid 83–1, which were then joined through a spatial union. Areas of each grid were calculated. There were slight differences between mapped area estimates and those calculated at the time of the study. We used the areas calculated at the time of the study, because they took into account small areas that were not surveyed due to missing stakes.

We also created shapefiles of the locations of all seal structures for each grid. To estimate structure locations, we first created points for the vertices of each grid (i.e., the intersection of grid lines at the corner of each block), which were manually labeled according to the naming system assigned during field activities. Coordinates were assigned to each point. These grid vertices (i.e. the corners of each survey block) constituted the fixed points that were used in the field to locate each structure based on angle and paces. The 'Bearing Distance To Line' tool (in the Data Management toolbox) was used to create lines that connected a corner of a block to the location of a seal structure, specifying the paced distance (in meters) and bearing (in degrees) from the known corner point. The end points of the lines were then extracted, representing the points of seal structures. Lastly, associated environmental measurements and attributes sampled in the field were appended to mapped seal structure points. Within each grid, we measured the distance between a structure and the nearest structure.

## Statistical analyses

We quantified differences in density, distance between structure types, and proportion of different lair types among grids and regions. Chi-squared tests were used to examine differences in proportions of lair types within the grids in each region. Analyses of how structure measurements differed among types of structures were examined using a series of one-way Analysis of Variance (ANOVA) tests for grids in Kotzebue Sound, Ledyard Bay and for all grids in both regions combined. Parametric assumptions were visually assessed using histograms and residual plots, and transformations were applied when the residuals exhibited nonconstant variance or nonnormality. We specifically estimated the differences in distances to the nearest other structure (square root-transformed), breathing hole diameter (log-transformed), area of seal structures (length x width, log-transformed), height of seal structures (log-transformed) as well as differences in snow depth, ice deformation (log-transformed), and ice relief among different subnivean structure types using ANOVAs. Differences among structures for each ANOVA were assessed using Tukey's Honestly Significant Difference tests. Because the second grid surveyed in Kotzebue Sound (83–2) during late April was already in late stages of melt and many lairs were collapsed and open when found, the structures within this grid were excluded from analyses of snow and ice characteristics by structure type. All statistical analyses

were conducted in the base packages of R [42]. Final model fits and validation of assumptions were assessed by investigating patterns in residuals, Q-Q plots, and leverage of residuals.

### Permits and approvals

No permits or approvals were required at the time of field data collection for this study, conducted in 1983 and 1984. The US Marine Mammal Protection Act protected ringed seals, but permits at that time were only required for invasive activities such as lethal collection of animals. No experiments with live animals were conducted, and no seals were captured, sampled, or handled. This study mapped and described non-permanent ringed seal structures excavated in snow, and all structures that were disturbed were reconstructed after they had been disturbed.

Field work was conducted on the ice covering unprotected marine waters. We acknowledge that our study areas constitute the traditional and contemporary territories stewarded by northwest Alaska's Iñupiaq People. Iñupiaq hunters from the communities of Deering, Alaska and Point Hope, Alaska supported and contributed to this work.

## Results

### Seal structures

We found a total of 490 ringed seal structures: 242 in Kotzebue Sound in 1983 and 248 in Ledyard Bay in 1984 (Fig 1, Table 2). Seven structures in 83–2 (3 pup lairs, 2 breathing holes, and 2 haul-out lairs) and one haul-out lair in 84–1 were determined to be off-grid and were not included in density calculations but were included in structure description statistics. Seal structures were found in higher overall density in the Ledyard Bay region during 1984 (8.06 structures/km$^2$) than in the Kotzebue Sound region in 1983 (7.07 structures/km$^2$). Three of the five Ledyard Bay grids had higher structure densities than either of the Kotzebue Sound grids. The lowest density in any grid of either study area (grid 84–5), was in a highly disturbed nearshore area in close proximity to a remote military site with regular vehicle traffic, heavy equipment, a

**Table 2. Summary characteristics of ringed seal structures located during on-ice surveys in Kotzebue Sound and Ledyard Bay during spring 1983 and 1984.**

| Grid | Area surveyed (km$^2$) | Number of structures | Density of structures (structures/km$^2$) | Mean distance (m) to nearest structure (s.d.) | Prop. breathing holes | Prop. haul-out lairs | Prop. pup lairs |
|---|---|---|---|---|---|---|---|
| *KOTZEBUE SOUND, 1983* | | | | | | | |
| **83–1** | 27.37 | 185 | 6.76 | 171.1 (102.3) | 0.34 | 0.46 | 0.20 |
| **83–2** | 5.87 | 57 [a] | 8.51 | 150.2 (122.1) | 0.37 | 0.23 | 0.40 |
| **TOTAL** | 33.24 | 242 | 7.07 | 166.2 (108.0) | 0.35 | 0.40 | 0.25 |
| *LEDYARD BAY, 1984* | | | | | | | |
| **84–1** | 10.92 | 93 [a] | 8.42 | 134.5 (118.0) | 0.30 | 0.37 | 0.33 |
| **84–2** | 5.32 | 36 | 6.76 | 211.1 (155.9) | 0.28 | 0.53 | 0.19 |
| **84–3** | 5.43 | 47 | 8.65 | 139.9 (121.7) | 0.09 | 0.56 | 0.36 |
| **84–4** | 5.56 | 60 | 10.79 | 135.6 (86.7) | 0.28 | 0.32 | 0.40 |
| **84–5** | 3.40 | 12 | 3.53 | 221.3 (130.7) | 0.25 | 0.42 | 0.33 |
| **TOTAL** | 30.63 | 248 | 8.63[b] | 151.1 (123.3) | 0.25 | 0.41 | 0.33 |
| *TOTAL ALL GRIDS* | *63.87* | *490* | *7.77[b]* | *158.6 (116.1)* | *0.30* | *0.41* | *0.29* |

[a] Off-grid detections of seal structures were excluded from density calculations (7 in grid 83–2 and 1 in grid 84–1).

[b] Grid 84–5 was highly disturbed relative to other grids (see Table 1), so is excluded from calculations of overall total density summaries for Ledyard Bay grids and all grids combined.

generator, and a nearby dump. For this reason, we excluded grid 84–5 from the overall density estimates for Ledyard Bay and for the two regions combined.

The distances between structures were generally shorter in grids with higher densities (Table 2), as would be expected. For example, grid 84–4 in Ledyard Bay had the highest density (10.79 structures/km$^2$) and shortest distance to nearest structure (135.6 ± 86.7 m) while grid 84–5 in Ledyard Bay had the lowest density (3.53 structures/km$^2$) and longest distance to nearest structure (221.3 ± 130.7 m). The composition of structure types was different in Kotzebue Sound ($\chi$ = 9.325, df = 2, $p$ = 0.009) than in Ledyard Bay ($\chi$ = 9.756, df = 2, $p$ = 0.008). The most common structures in both regions were haul-out lairs (0.41 for both regions combined), but the proportion of pup lairs was higher in Ledyard Bay (0.33) than in Kotzebue Sound (0.25). Within Kotzebue Sound, the proportion of pup lairs in grid 83–2 (0.40) was twice that in grid 83–1 (0.20) and matched the highest proportion of pup lairs among grids in Ledyard Bay (in grid 84–4). Simple breathing holes were more common in Kotzebue Sound than in Ledyard Bay (Table 2).

Structure characteristics differed among the types of seal structures (Fig 3; Table 3). The distance from a structure to the nearest other structure varied significantly among structure types in both regions as well as regions combined (Kotzebue Sound: F = 11.41, df = 2, $p$ < 0.0001; Ledyard Bay: F = 6.98, df = 2, $p$ = 0.001; both areas combined: F = 18.22, df = 2, < 0.0001). The shortest distances to the nearest other structure were found for pup lairs (overall mean distance = 126.5 m). The diameter of breathing holes varied significantly among structure types in both regions separately (Kotzebue Sound: F = 19.15, df = 2, $p$ < 0.0001; Ledyard Bay: F = 10.65, df = 2, $p$ = 0.00004) and combined (F = 26.48, df = 2, < 0.0001). Within Kotzebue Sound, breathing holes were significantly larger in grid 83–2 than in grid 83–1 (F = 8.17, df = 1, $p$ = 0.005). Overall, simple breathing holes were smallest (overall mean diameter = 29.8 cm), and pup lairs had the largest breathing holes, which also provided access into the lair (overall mean diameter = 48.2 cm). Structure area varied significantly between haul-out and pup lairs in both regions separately (Kotzebue Sound: F = 81.7, df = 1, $p$ < 0.0001; Ledyard Bay: F = 131, df = 1, $p$ < 0.0001) and combined (F = 214.5, df = 1, p < 0.0001). Pup lairs were larger than haul-out lairs (overall mean area = 4.1 m$^2$ and 1.8 m$^2$, respectively) and had longer mean lengths and widths (Table 3). The largest area of a single structure (15.1 m$^2$) was measured at a pup lair in grid 83–2. It contained 9 chambers and was extensively tunneled, measuring 485 cm long, 311 cm wide, and 30 cm high. Height also varied significantly between haul-out and pup lairs in both regions separately (Kotzebue Sound: F = 4.3, df = 1, $p$ = 0.0389; Ledyard Bay: F = 6.754, df = 1, $p$ = 0.0102) and combined (F = 13.06, df = 1, p < 0.0001). Pup lairs were higher than haul-out lairs (overall mean height = 32.6 cm and 30.1 cm, respectively). For all metrics, structures were larger in Kotzebue Sound than in Ledyard Bay.

## Snow and ice characteristics

Snow depth varied significantly among structure types (Kotzebue Sound: F = 53.85, df = 2, $p$ <0.0001; Ledyard Bay: F = 48.23, df = 2, $p$ < 0.0001; Fig 4). In general, snow depth was deepest at pup lairs, although not significantly different from haul-out lairs in Kotzebue Sound ($p$ = 0.085). Mean snow depth was deeper at structures in Kotzebue Sound than those in Ledyard Bay (66.2 and 58.3 cm, respectively), but deep snow outliers were more common in Ledyard Bay where the deepest snow depth (183 cm) was at a haul-out lair noted as "in a huge [snow] drift" (Table 4). Minimum snow depth for haul-out lairs was 32 cm in Ledyard Bay and 35 cm in Kotzebue Sound. Minimum snow depth for pup lairs was 37 cm in Ledyard Bay, although the roof of this lair was open; height measured only 18 cm at this lair. For intact pup lairs that included direct evidence of a pup, minimum snow depth was 47 cm in Ledyard Bay

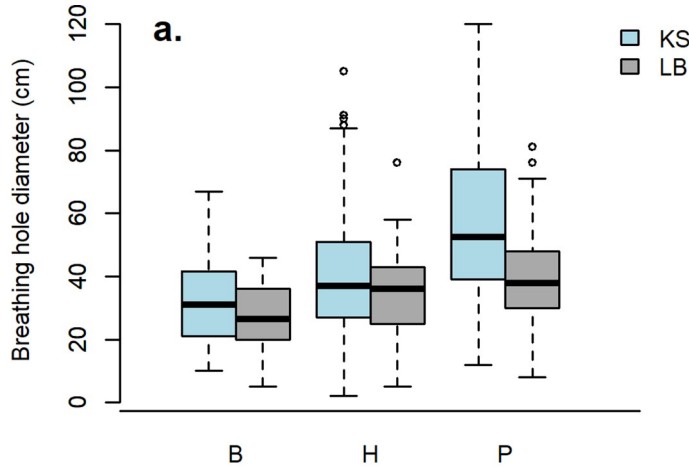

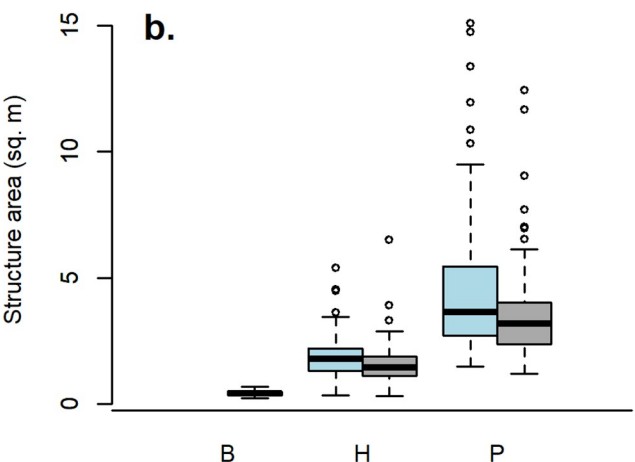

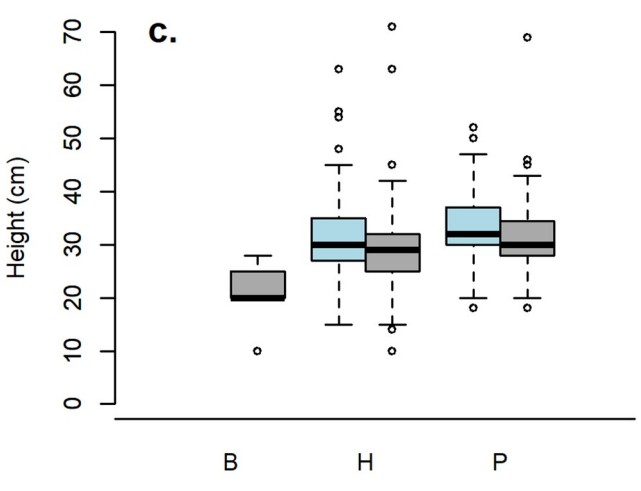

**Fig 3. Boxplots summarizing dimensions of seal structures (B = breathing hole, H = haul-out lair, P = pup lair) in Kotzebue Sound (KS) and Ledyard Bay (LB).** Dimensions include: a) diameter of breathing holes, b) area of structures, and c) height of structures. Boxes correspond to the lower and upper quartiles, and black lines indicate median values. Whiskers and open dots indicate extreme values and outliers, respectively.

and 58 cm in Kotzebue Sound. Ice deformation was not significantly different among structure types in Kotzebue Sound (F = 1.92, df = 2, $p$ = 0.149), but in Ledyard Bay (F = 5.98, df = 2, $p$ = 0.0029) it differed between breathing holes and lairs ($p \leq 0.005$) but not between haul-out and pup lairs ($p$ = 0.947). Mean deformation was similar in both regions (26% in Kotzebue Sound, 29% in Ledyard Bay). Ice relief was not significantly different among structure types within each region (Kotzebue Sound: F = 2.531, df = 2, $p$ = 0.0825; Ledyard Bay: F = 2.363, df = 2, $p$ = 0.0963). Mean ice relief height was higher in Kotzebue Sound (79.9 cm) than in Ledyard Bay (65.7 cm).

## Discussion

### Characteristics of seal structures

All lairs begin as breathing holes. Some breathing holes are excavated into simple single-chambered lairs once enough drifted snow has accumulated over them, and some of those are later enlarged by pups to include multiple chambers. Across the Arctic, pup lairs are larger than haul-out lairs [8,34,43]. In this study, the mean lengths and widths of pup lairs were more than 1.5 times greater than those of simple lairs. There is geographic variation in reported lair size (Table 5). Haul-out and pup lairs sampled in the eastern Beaufort Sea [8] were on average 20–

**Table 3. Mean (± s.d.), minimum-maximum, and *n* sample size descriptive statistics for each type of ringed seal structure (B = simple breathing hole, H = haul-out lair, P = pup lair) in Kotzebue Sound and Ledyard Bay regions of the Chukchi Sea.**

| | Kotzebue Sound | | | Ledyard Bay | | | Regions combined | | |
|---|---|---|---|---|---|---|---|---|---|
| | **B** | **H** | **P** | **B** | **H** | **P** | **B** | **H** | **P** |
| Length (cm) | - | 162.4 (43.6) 65–303 97 | 282.9 (107.3) 150–641 56 | - | 151.0 (46.3) 58–427 99 | 251.7 (74.8) 137–566 80 | - | 156.6 (45.2) 58–427 196 | 264.5 (90.6) 137–641 136 |
| Width (cm) | - | 112.0 (29.8) 40–224 97 | 161.9 (55.2) 73–351 56 | - | 99.4 (24.9) 33–183 99 | 138.4 (45.7) 56–343 80 | - | 105.7 (28.1) 33–224 196 | 148.1 (51.0) 56–351 136 |
| Height (cm) | - | 31.5 (8.1) 15–63 88 | 34.2 (7.8) 18–52 53 | - | 28.9 (8.5) 10–71 96 | 31.6 (7.5) 18–69 76 | - | 30.1 (8.4) 10–71 184 | 32.6 (7.7) 18–69 129 |
| Area of lair (m²) | - | 1.9 (0.9) 0.3–5.4 97 | 4.9 (3.3) 1.5–15.1 56 | - | 1.6 (0.8) 0.3–6.5 99 | 3.6 (2.1) 1.2–12.5 80 | - | 1.7 (0.9) 0.3–6.5 196 | 4.1 (2.7) 1.2–15.1 136 |
| Diameter breathing hole (cm) | 31.6 (13.4) 10–67 75 | 41.6 (21.1) 2–105 86 | 58.2 (27.3) 12–120 52 | 27.5 (9.5) 5–46 60 | 34.9 (13.1) 5–76 85 | 39.5 (13.7) 8–81 59 | 29.8 (11.9) 5–67 135 | 38.3 (17.9) 2–105 171 | 48.2 (23.1) 8–120 111 |
| Distance to nearest other structure (m) | 203.6 (102.7) 2–423 84 | 159.6 (103.5) 0–416 97 | 123.2 (107.6) 0–394 59 | 195.1 (121.2) 3–571 62 | 144.2 (130.0) 1–743 102 | 128.9 (109.3) 1–474 82 | 200.0 (110.6) 2–571 146 | 151.7 (117.8) 0–743 199 | 126.6 (108.2) 0–474 141 |

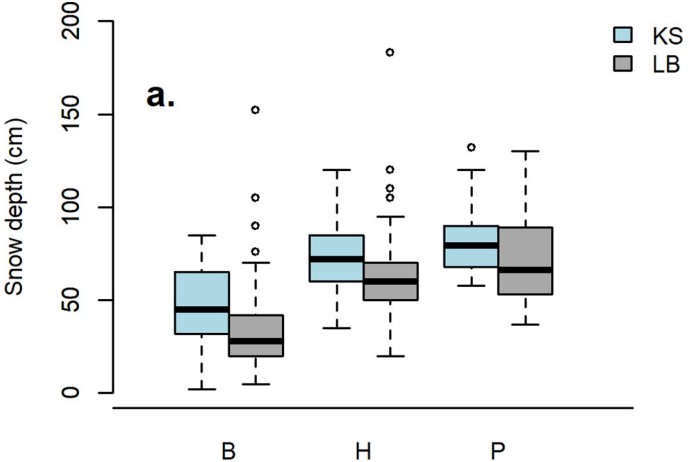

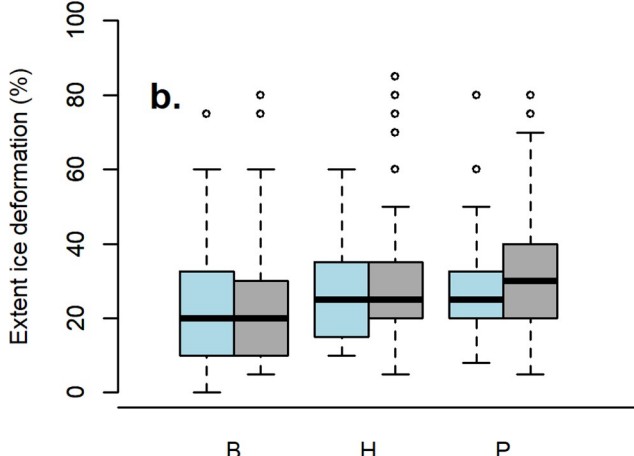

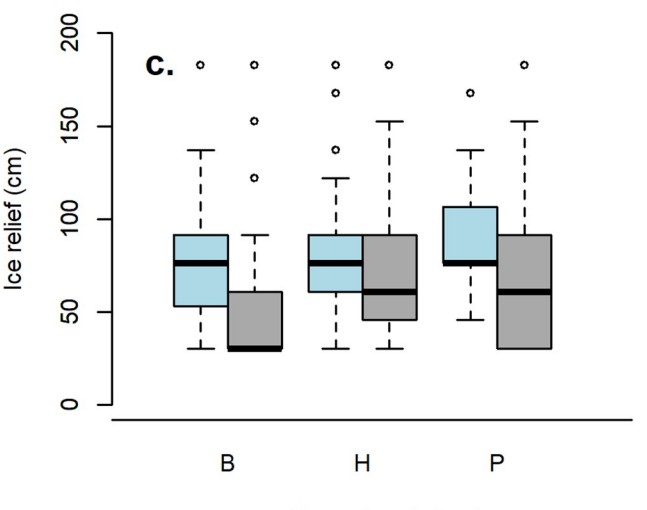

**Fig 4. Boxplots summarizing environmental conditions at seal structures (B = breathing hole, H = haul-out lair, P = pup lair) in Kotzebue Sound (KS) and Ledyard Bay (LB).** Environmental conditions include: a) snow depth at the structure, b) extent of ice deformation, and c) ice relief height. Boxes correspond to the lower and upper quartiles, and black lines indicate median values. Whiskers and open dots indicate extreme values and outliers, respectively.

30% larger than those measured in this study. Haul-out lairs in Svalbard [34] were also larger (mean = 2.4 m$^2$, breeding males only) than haul-out lairs in this study (means of 1.9 m$^2$ in Kotzebue Sound and 1.6 m$^2$ in Ledyard Bay, not differentiated by sex of the occupant). Pup lairs in Svalbard were intermediate in size (4.3 m$^2$) between our two Alaska study areas (Kotzebue Sound = 4.9 m$^2$; Ledyard Bay = 3.6 m$^2$). Ringed seal lair heights were similar (31–34 cm) in all regions across the Arctic for which there are published data (Table 5). We found seven structures that we classified as "breathing holes" because they were too small (length 46–86 cm, width 41–79 cm) to accommodate a hauled out adult seal. We suspect they were in the process of becoming lairs, although it also possible they could have been small pup escape lairs [8].

Seal size likely affects the size of lairs. Recent comparative analyses indicate substantial variation in body size across the range of ringed seals suggestive of different morphs, with seals in Alaska among the smallest [49]. Ringed seals in both the eastern Beaufort Sea region and Svalbard regions were longer than those in the eastern Chukchi Sea. Mean standard length of adult ringed seals was 124.7 cm in Amundsen Gulf [45], 128.9 cm in Svalbard [34] and 114.6 cm in Alaska [50]. The mean length of adult ringed seals in Alaska was consistent with the mean size of haul-out lairs measured in this study (157 cm x 106 cm). Lair height was less variable among studies, which presumably corresponds to seal girth. Although Alaska ringed seals were smaller than those in some other areas [49,50], a substantial difference in seal girth would not substantially affect lair height. Differences in methods among studies may have impacted lair measurements, especially for complex, multi-chambered pup lairs. In our study, lair measurements were made through a hole that was just large enough to accommodate the researcher's head and shoulders whereas lairs in other studies were often completely excavated and could be fully viewed when measured. Snow characteristics, such as depth and density, also vary and could affect lair size.

Breathing holes were larger when they served as access holes to pup and haul-out lairs than when they were used simply for breathing in both this and other studies (Table 5). Breathing holes in pup lairs were on average larger than holes in haul-out lairs. Females nursing pups likely enter lairs to attend their pups more frequently than non-nursing seals, and the increased use may create a somewhat larger hole. Parameters used to classify lairs likely affects whether differences are seen among lair types. In this study, all lairs with more than one

**Table 4. Mean (± s.d.), minimum-maximum, and *n* sample size descriptive statistics for environmental conditions at each type of ringed seal structure (B = simple breathing hole, P = pup lair, H = haul-out lair) in Kotzebue Sound and Ledyard Bay regions of the Chukchi Sea.**

| | Kotzebue Sound | | | Ledyard Bay | | |
|---|---|---|---|---|---|---|
| | B | H | P | B | H | P |
| Snow depth (cm) | 46.5 (19.6) 2–85 62 | 74.0 (18.9) 35–120 84 | 81.9 (16.9) 58–132 36 | 35.3 (24.8) 5–152 61 | 61.6 (20.5) 32–183 100 | 71.7 (22.2) 37–130 80 |
| Extent ice deformation (%) within 200 m | 23.0 (15.8) 0–75 63 | 26.7 (14.3) 10–60 83 | 28.1 (15.8) 8–80 36 | 24.7 (19.9) 5–80 62 | 29.9 (17.7) 5–85 101 | 30.7 (17.3) 5–80 81 |
| Ice relief height (cm) within 200 m | 71 (34) 31–183 63 | 80 (29) 31–183 84 | 88 (26) 46–168 36 | 58 (41) 31–183 62 | 67 (32) 31–183 102 | 69 (35) 31–183 82 |

**Table 5. Comparisons of mean (minimum-maximum) measurements of ringed seal lair dimensions and environmental conditions, based on surveys conducted on the landfast ice during the pupping season and using trained dogs.**

| | Kotzebue Sound, Alaska 66°N Apr 1983 | Ledyard Bay, Alaska 69°N Apr-May 1984 | Beaufort Sea, Alaska 70°N Mar-Apr 1982–1983 | Amundsen Gulf, Canada 70–72°N Mar-Apr 1971–1983 | Barrow Strait, Canada 74–75°N Mar-Jun 1975, 1984–1986 | SE Baffin Island, Canada 62–64°N Mar-Jun 1991–1993 | White Sea, Russia 64–67°N Apr-May 1972–1974 | Svalbard, Norway 78–79°N Feb-Apr 1984 |
|---|---|---|---|---|---|---|---|---|
| **Percent structures** | | | | | | | | |
| Breathing hole | 35 | 25 | 35 | 28 (5–41) | 20–40 | 24 | 77 | |
| Haul-out lair | 40 | 41 | 61 | 53 (33–71) | | 52 | 14 | |
| Pup lair | 25 | 33 | 4 | 19 (0–60) | | 24 | 9[a] | |
| **Breathing hole diameter (cm)** | | | | | | | | |
| Breathing hole | 32 (10–67) | 28 (5–46) | | 40 | | 46 | 30–40 | |
| Haul-out lair | 42 (2–105) | 35 (5–76) | | 40 | | 51 | 40–50 | |
| Pup lair | 58 (12–120) | 40 (8–81) | | | | 49 | | |
| **Lair height (cm)** | | | | | | | | |
| Haul-out lair | 32 (15–63) | 29 (10–71) | | 31 (10–60) | | 29 | 40–60 | 31 (20–60) |
| Pup lair | 34 (18–52) | 32 (18–69) | | 31 (16–65) | | 32 | | 31 (15–40) |
| **Lair length (cm)** | | | | | | | | |
| Haul-out lair | 162 (65–303) | 151 (58–427) | | 197 (30–400) | | 166 | 150–300 | |
| Pup lair | 283 (15–641) | 252 (137–566) | | 355 (60–900) | | 152 | | |
| **Lair width (cm)** | | | | | | | | |
| Haul-out lair | 112 (40–224) | 99 (33–183) | | 138 (30–400) | | 116 | 60–80 | |
| Pup lair | 162 (73–351) | 138 (56–343) | | 228 (30–500) | | 152 | | |
| **Lair area (m²)** | | | | | | | | |
| Haul-out lair | 1.9 (0.3–5.4) | 1.6 (0.3–6.5) | | 2.7–3.8 | 4.6–8.1 | 1.9 | 3.5 | 2.4 |
| Pup lair | 4.9 (1.5–15.1) | 3.6 (1.2–12.5) | | | | | | (1.2–13.3) |
| **Distance to nearest structure (m)** | | | | | | | | |
| Breathing hole | 204 (2–422) | 195 (3–571) | | 233 (27–563) | | 290 | | |
| Haul-out lair | 160 (0–416) | 144 (1–743) | | 124 (25–400) | | 215 | | |
| Pup lair | 123 (0–394) | 129 (1–474) | | | | 27 | | |
| **Snow depth (cm)** | | | | | | | | |
| Breathing hole | 47 (2–85) | 35 (5–152) | | 60 (20–150) | 37–53 | 43 | | 91 (55–145) |
| Haul-out lair | 74 (35–120) | 62 (32–183) | | 65 (25–150) | 50–69 | 65 | | 89 (65–130) |
| Pup lair | 82 (58–132) | 72 (47[b]–130) | | | 51–79 | 71 | | |
| **Structure density (structures/km²)** | 6.8–85 | 6.8–10.8 | 0.8[c] | 0.5–1.3 | 4.8–7.9 | 8.3–12.3 | 8.3–27.0 | 5.5 |
| **Source** | This study | This study | [44] | [8,45] | [46,47] | [43] | [48] | [23,34,40] |

[a] Researchers included flat ice and areas with little snow cover in their survey. Within areas of adequate snow cover and deformation, up to 33% of all structures were pupping lairs.

[b] Minimum snow depth for an unaltered pup lair that contained evidence of a pup.

[c] Birth lairs only.

chamber were assumed to be used by females with pups (and often confirmed by observing evidence of pup use), while in some studies "pup lairs" were divided into multiple types of structures, including very small lairs that were apparently used only by pups [8].

Distances between ringed seal structures vary among studies [8,34,36]. We noted when structures were in close proximity during our field work, but distances to the nearest structure were calculated after field work was completed using GIS techniques. The overall mean distance to the nearest structure was 159 m (range = 0–743 m), with considerable variation among grids (Table 2). Pup lairs were located closer to other structures (mean = 127 m, range = 0–474 m) than haul-out lairs (mean = 152 m, range = 0–743 m) or breathing holes (mean = 200 m, range = 2–571 m). Smith and Stirling [8] described "lair complexes" as clusters

of pup lairs and other nearby lairs and breathing holes that presumably provide alternative haul-out or breathing locations as a means of predator avoidance. The mean distance between structures they considered to be associated, based simply on proximity, was 23 m. Other studies have reported that pup lairs are often found close to other subnivean structures, consistent with the concept that adult females occupy a birth lair complex within a set area often associated with a large snow drift or downwind of a prominent ice feature [34,43]. The first study of lair use by radio-tagged seals, conducted in the Beaufort Sea, confirmed that seals used multiple lairs, and also that the same lair was sometimes used by more than one seal [36]. However, distances between lairs were substantial, ranging from 300–3448 m. The mean distance between lairs used by individual males was 1997 m, compared to 634 m for females. One female located in a birth lair used four different lairs. In Svalbard, tagged pups that were still nursing used an average of nine structures (breathing holes plus lairs) per pup, located up to 1400 m apart [51]. We did not instrument seals in this study and cannot assess the number of structures used by individual seals or the distances between an individual seal's structures. It is likely that the number of structures used by individual seals, as well as the distance among associated structures, vary according to local seal densities, sex and reproductive status of the seal, prey availability, as well as the distribution and characteristics of local ice and snow conditions.

Pup lairs are characterized by greater ice deformation and deeper snow than other structure types [43,52, this study, 46]. Across study areas, mean snow depths at pup lairs ranged 65–89 cm, 60–91 cm at haul-out lairs, and 35–47 cm at breathing holes (Table 5). It is not possible to construct lairs on flat, windswept ice where there is little or no snow. However, snow accumulates in areas with pressure ridges or jumbled ice, especially on the downwind side, and makes it possible for seals to excavate lairs in this dense wind-packed snow, even in areas where snowfall is limited. In this study, lairs were located on ice with more deformation than were breathing holes. It is possible that simple breathing holes located under more deformed ice could be harder for dogs to detect, since breathing holes may have less seal scent than lairs where seals' whole bodies contact the cavity wall. However, previous work indicated no bias in the types of structures each dog located [41], and breathing holes may actually be easier to detect because they are often found along linear features and flat ice with less snow cover [e.g., 43,46]. Grids that were generally characterized by higher overall ice relief and/or diversity of ice topography (e.g. grids 83–2 and 84–4) had higher densities of structures than flatter grids with many refrozen leads, flat ice pans, or fairly uniform conditions (e.g. grids 83–1 and 84–1).

## Composition of seal structures

The proportions of ringed seal structure types in the two regions we studied were generally similar. However, the proportions of structure types were different among the individual grids within regions, with the proportion of pup lairs differing by a factor of two among some grids. In smaller grids, the amount of extensive flat ice versus deformed ice with more relief substantially affected snow accumulation and therefore the amount of habitat suitable for lairs.

The composition of seal structures varies considerably across its range (Table 5). As in our study areas, the types of structures found was affected by the size and ice characteristics of the study area. For example, 9% of structures were pup lairs in areas containing flat ice not suitable for lairs in a study in the White Sea [48]. In areas with adequate snow cover and deformation, 33% of all structures were pup lairs. Similarly, in Svalbard, while the density of pup lairs in prime breeding habitat was comparable to structure densities across Canada and Alaska, only 1.4% of all Svalbard sea ice was suitable habitat [23]. It is the overall amount of suitable ice and adequate snow, mediated by other factors such as prey resources, that ultimately determines

the number and type of seal structures and breeding seals within a region. Quantitative information about suitable habitat was not available in the 1980s and 1990s when most studies of seal breeding habitat were conducted.

## Regional abundance of seal structures

Few other studies of ringed seal structures and pupping habitat have provided detailed information about the density of structures over relatively large areas. When density estimates were provided, they were generally extrapolated from one-time searches as estimates of the width of the search area covered by dogs (usually estimated to be 200 m) multiplied by the linear distance searched. In this study, dogs searched survey grids along lines spaced such that they could thoroughly survey the entire grid. Most lines were searched at least twice, and some up to four times, so that detection rates could be evaluated. In total, our survey grids covered nearly 64 km$^2$. Structure density varied between regions. Total density was about 20% higher in the Ledyard Bay region in 1984 than in Kotzebue Sound in 1983. Although suitable conditions for lair construction (greater ice deformation and deeper snow) occurred in both regions, we did not quantify the overall availability of suitable lair habitat. Local conditions, in particular snow depth, affect the distribution of lairs. With a single year of surveys in each region, it is not possible to distinguish differences in the availability of suitable breeding habitat between regions from inter-annual differences. Density differences can also be due to or augmented by factors such as prey availability.

Aerial surveys of ringed seals on fast ice along the Alaska coast in 1985–1987 indicated substantial inter-annual variation in seal density, both within and among the survey sectors that included Kotzebue Sound and Ledyard Bay [38]. Between-year density differences within sectors ranged from +90% to -48%. In 1985, seal densities were higher in Ledyard Bay, in the next they were lower, and in 1987 they were similar. In the White Sea, Lukin and Potelov [48] noted a 30% reduction in structure density between 1972 and 1973. Smith and Stirling [8] found 1 pup lair per 6 minutes of searching in 1973 compared to 1 per 64 minutes the following year. On-ice studies of seal lairs and aerial surveys of hauled out seals both demonstrate that ringed seal habitat is dynamic and highly variable on both temporal and spatial scales.

## Historical measurements in the context of contemporary changes in ice and snow

To some extent ringed seals can mediate inter-annual variability and variations in environmental conditions [28,39], yet climate change has imposed unidirectional and rapid changes in ice and snow since the time of our surveys in 1983 and 1984. Such changes affect the quantity and quality of available ringed seal habitat. The extent of landfast ice and the duration of seasonal cover in the Chukchi Sea region have significantly declined, particularly as freeze-up occurs later in the autumn and break-up commences earlier in the spring [53]. Until the early 21$^{st}$ century, Kotzebue Sound was consistently covered with stable landfast ice during at least January-April, extending in a continuous expanse along the shore and seaward to the entrance of the Sound [53]. Landfast ice has not completely covered Kotzebue Sound during winter for approximately the last decade [54]. Only relatively narrow margins of landfast ice attached to the periphery were formed in several recent years. The open water season along the Chukchi Sea coast has lengthened significantly since the late 1970s, although regions like Kotzebue Sound show little net change and instead are characterized by substantial and increased inter-annual variability [55].

Snow depth on sea ice has also declined in the Chukchi Sea with shorter ice-covered seasons in recent decades [56,57]. However, widespread spatial and temporal measurements are

limited, and only a few areas have been repeatedly monitored. Indeed, climate-related changes in snow accumulation on sea ice have been challenging to track over time and across broad regions of the Arctic, because there were not programs for routine acquisition of measurements. It is only recently that remotely-sensed measurements have become available [58]. Future projections suggest that precipitation will increasingly fall as rain rather than snow, shortened sea ice-covered seasons with diminished spatial extent will result in decreased periods and areas for snow accumulation, and warmer air temperatures could result in earlier melt [30,32,59]. A recent study also indicated that the weight of accumulated snow on top of unusually thin landfast ice can contribute to widespread surface flooding [54]. If such conditions coincide with the pupping period, lairs used by pups still in lanugo may flood and compromise the pup's ability to stay warm. Much work remains to overcome the limited records and information gaps that are needed to better understand present and future depths, distribution, and variability of snow accumulation on sea ice [56].

Despite the importance of ice deformation and snow accumulation for lairs to protect pups from exposure and predation [9,23,48,60], it is difficult to quantify the minimum snow depths and ice topography that are required, much less optimal, for pup survival. We found that areas of flat ice with little or no snow cover generally have simple breathing holes but no subnivean structures. Minimum snow depth measured at a lair classified as a pup lair was 37 cm (range 37–132 cm), though this lair was open to the air and had been modified to a breathing hole. Minimum snow depth at a lair that was unaltered and with actual signs of a pup was 47 cm. Minimum snow depths measured in our study are similar to values reported for Canada and Svalbard in the 1980s (Table 5) and consistent with the statement by Smith and Lydersen [23] that seals require a minimum of 45 to 130 cm of snow cover to excavate birth lairs. The minimum snow depth for pup lairs was 25 cm in the eastern Beaufort Sea in the early 1970s [8]. Ferguson et al. [26] estimated an inflection point between snow depth and recruitment at 32 cm snow depth, yet cautioned against using this value for management purposes. These values were for snow accumulation on flat ice, not snow depths in drifts where lairs are excavated. Modeling exercises have used 20 cm snow depth measured on flat ice as a critical threshold for the successful creation of pup lairs [32], below which 100% pup mortality was assumed to occur [33]. Ultimately, minimum snow depth required for a lair is not the same as minimum snow depth on flat ice, and it is difficult to relate the two in a quantitative manner.

Snow depth at a lair is the sum of the depth of snow forming the roof plus lair height, which is a function of seal size. Accordingly, a very small seal can excavate a lair in less snow than a larger seal. Physical snow characteristics (e.g. water equivalent, color, temperature) impact the structural integrity of a lair roof, and therefore the required minimum snow depth. If snow depth is measured late in the spring when the weather is warming and the snow has begun to melt and condense, measured minimum snow depths may not reflect conditions at the time the lair was constructed. Mean snow depth at pup lairs may be a better measure of suitable habitat.

Lairs excavated in deeper snow afford greater protection to seals. Past studies indicate polar bear predation of ringed seals was less successful and fewer attempts on known lairs occurred in areas of deeper snow [43,46]. Lairs with shallow snow depth may not provide sufficient insulation and are more susceptible to early collapse or melt. For example, frozen pups were found in lairs with snow roofs of 5–10 cm in the White Sea [48]. Exposed newborn pups have been found after unusual warm and rain events prematurely collapsed lairs off southeastern Baffin Island [60]. Premature collapse would particularly jeopardize pups that are still in lanugo through thermoregulatory stress from exposure to wind and cold, especially when they are wet [9]. Overall, the timing of snowmelt, integrity of a lair, and a pup's ability to insulate

itself are important factors for understanding what minimal snow depths are sufficient for pup survival.

Lair melt and ceiling collapse are natural parts of the typical spring melt process. We excluded grid 83–2 from our analysis of environmental conditions because of the advanced stage of melt in late April 1983 that was not comparable to the other grids surveyed earlier in the melt process. The range in partially melted snow depths was 5–77 cm (mean = 49.7 cm) at pup lairs, compared to 58–132 cm (mean = 81.9 cm) on the other Kotzebue Sound grid (83–1) that was surveyed before melting began. In grid 83–2, 30% of 23 pup lairs were collapsed. None showed indications of predation, but the advanced stage of snow melt and collapse of lair ceilings likely obscured any tracks and/or entry holes dug by predators. Collapsed lairs occurred at a much lower rate on other grids that were surveyed earlier (1–2 collapsed haul-out lairs in each of the five Ledyard Bay study grids and two collapsed pup lairs in grid 84–4). Many of these either had signs of fox entry or were surveyed relatively late in the spring (30 April-7 May 1984). The advanced stage of melt in grid 83–2 may have contributed an upward bias to the size of breathing holes, which were about 20% larger than breathing holes measured in grid 83–1 (83–1 mean diameter = 40.1 cm, range = 2–120 cm; 83–2 mean diameter = 48.5 cm, range = 12–109 cm). The margins of exposed holes may have enlarged more rapidly once lairs were open to the air. Alternatively, or in addition, breathing holes in grid 83–2 that were surveyed later in the season may have been larger due to erosion of the margins by frequent use. Overall, breathing holes in both Kotzebue Sound grids were larger than breathing holes in Ledyard Bay grids (mean diameter = 34.0 cm, range = 5–81 cm).

The construction and use of lairs by three other ringed seal conspecifics differs somewhat from *P.h. hispida*, the subspecies described in this and other studies across the US, Canadian, Russian, and Svalbard Arctic. Two other ringed seal conspecifics, Saimaa seals (*Pusa hispida saimensis*) and Lake Ladoga seals (*P. h. lagodensis*) have adapted to living in large lakes that generally have less ice deformation and less snow accumulation on ice. These lake seals build lairs adjacent to nearshore hummocks or along the shoreline where snow accumulates against the land [61,62]. Saimaa seals sometimes have pups in snow depths of 25 cm or less, but the "lairs" have no roofs and are open on the top [63]. Another conspecific, the Okhotsk seal (*P. h. ochotensis)* is also apparently able to care for its pups without building lairs in the snow [64]. Females with pups in lanugo were seen near hummocks with snow in fractured ice.

A common assumption is that climate change is negatively impacting the ice and snow conditions in which ringed seals bear and nurture their young [33]. However, ringed seals have been considered to be the least sensitive to climate change of the eleven Arctic and sub-Arctic marine mammal species [65] as well as somewhat resilient based on their generalist diet, large numbers, and wide geographic range [66]. Across their range, ringed seals, as well as their primary predator [67,68], may have different responses to environmental changes. Ringed seals sampled in northern Alaska during 2003–2011 had thicker blubber, grew faster, matured earlier and had a higher proportion of pups in the harvest than those sampled during 1975–1984 [7]. This suggests that ringed seals in waters adjacent to Alaska had not been negatively impacted by changing environmental conditions. Body condition in the Amundsen Gulf was highly variable during both the 1970s and 1990s, but ovulation rates were higher in the 1990s when the open water period was a month longer (similar to Alaska). In 1998, breakup in Amundsen Gulf was six weeks earlier than mean date and prey appeared to be abundant and available to all age classes of seals [69,70]. Seals harvested that year were in good condition and ovulation rates were high. Pups, however, were in poor condition and starveling pups were commonly observed. The early break up presumably interrupted the nursing period and likely impacted the survival of unweaned pups [69]. When tracked more broadly from 1971–1978 and 1992–2019, there was a decline in body condition, and reproductive failures (defined as

50% of the multiparous females failing to ovulate) occurred in three of the 36 years but rebounded within 1–3 years [71]. Overall, declining trends in body condition and episodic reproductive failures could not be fully explained by local environmental conditions [71], and predicted impacts of the changing environment are still not fully understood. Ice loss and associated potential impacts to pupping habitat have also not been uniform across areas where ringed seals occur. Ringed seals in regions at the southern boundaries of their geographic range may be disproportionately affected [29]. Similarly, sea ice loss can be punctuated with years of extreme change that may play outsized roles in population-level ringed seal responses [25].

## Conclusions

Snow depth, modified by sea ice complexity and snow characteristics, represents a continuum of conditions required by ringed seals for the creation of lairs with sufficient strength and longevity to protect pups from predation and weather. None of the historical studies, including ours, were designed to determine minimum snow depths that ringed seals require to establish lairs and ultimately produce pups to the weaning stage. Additional data are required to evaluate contemporary environmental conditions, particularly snow depth and when the ice platform melts, in areas where ringed seals bear their pups in the context of historical records, and to evaluate the implications of any changes for reproductive success.

Recent studies suggest that the Bering and southern Chukchi seas are undergoing rapid ecological transition [72], which could affect ringed seal breeding habitat. Current measurements of snow depth and ice relief, especially if done in conjunction with estimates of reproductive success, may help reveal whether ringed seals in the region are already experiencing marginal habitat conditions for successful pupping. Further advances in the ability to estimate widespread snow depths on sea ice [56] could additionally support our ability to understand how habitat critical for seal pups is changing. The data presented in this study for ringed seals in Alaska, together with similar data from other studies across the Arctic, provide valuable baseline information about ringed seal habitat that predates recent climate warming and associated reductions in snow accumulation and sea ice formation and longevity. In combination with information about ringed seal body condition, demographics, pup production and current environmental conditions, it will facilitate a better understanding of how a warming climate may impact ringed seals in the future.

## Supporting information

**S1 Fig.**
(JPG)

## Acknowledgments

This analysis builds on a project entitled "Winter Ecology of Ringed Seals in Alaska" conducted by the Alaska Department of Fish & Game. Many people contributed to that project. Thomas G. Smith provided the opportunity for and initial direction during the training of our "seal dogs." George Lapiene of the NOAA/OCSEAP office was persistent in his efforts to arrange for the complex logistical requirements of the project. Able field assistance and appreciated companionship were also provided by Sue Hills, Robert R. Nelson, Taylor Moto (Deering), Ann M. Adams and Rex Tuzroyluk, Jr. (Point Hope). Lloyd Lowry contributed to this project as a thoughtful organizer, participant in field work and editorial reviewer. Personnel of the US Air Force facility at Cape Lisburne were gracious hosts and generous in allowing the

all-too-frequent use of their repair shop during the Ledyard Bay study. We are grateful to Peter Boveng and Justin Crawford for their helpful reviews and suggestions that greatly improved this manuscript.

## Author Contributions

**Conceptualization:** Kathryn J. Frost, John J. Burns.

**Data curation:** Donna D. W. Hauser, Kathryn J. Frost.

**Formal analysis:** Donna D. W. Hauser, Kathryn J. Frost.

**Funding acquisition:** Donna D. W. Hauser, Kathryn J. Frost, John J. Burns.

**Investigation:** Kathryn J. Frost, John J. Burns.

**Methodology:** Kathryn J. Frost, John J. Burns.

**Project administration:** Donna D. W. Hauser, Kathryn J. Frost, John J. Burns.

**Resources:** John J. Burns.

**Supervision:** John J. Burns.

**Validation:** Kathryn J. Frost.

**Visualization:** Donna D. W. Hauser.

**Writing – original draft:** Donna D. W. Hauser, Kathryn J. Frost.

**Writing – review & editing:** Donna D. W. Hauser, Kathryn J. Frost, John J. Burns.

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
