## [Decision Letter · Decision Letter 0]

29 Oct 2021

PONE-D-21-28218Ringed seal (Pusa hispida) breeding habitat on the landfast ice in northwest Alaska during spring 1983 and 1984PLOS ONE

Dear Dr. Hauser,

Thank you for submitting your manuscript to PLOS ONE. After careful consideration, we feel that it has merit but does not fully meet PLOS ONE’s publication criteria as it currently stands. Therefore, we invite you to submit a revised version of the manuscript that addresses the points raised during the review process.

This manuscript has now been reviewed by two researchers who are experts on ringed seal ecology. Both were very positive about this manuscript, and only provided relatively minor suggestions for improving the manuscript. Please address all of these comments in your revision.

We look forward to receiving your revised manuscript.

Kind regards,

William David Halliday, Ph.D.

Academic Editor

PLOS ONE

Journal Requirements:

"We are grateful to the North Pacific Research Board for funding this analysis (Grant No. 1811). This analysis builds on a project entitled “Winter Ecology of Ringed Seals in Alaska” conducted by the Alaska Department of Fish & Game and funded by the Outer Continental Shelf Environmental Assessment Program as RU 232 under contract NA-81-RAC-00045. Many people contributed to that project"

"Analysis of these data was funded by the North Pacific Research Board, Grant No. 1811, to D.D.W.H. and K.J.F. Data were originally collected with funding from the Outer Continental Shelf Environmental Assessment Program, contract NA-81-RAC-00045 to J.J.B. and K.J.F. The funders had no role in study design, data collection and analysis, decision to publish, or preparation of the manuscript"

3. We note that you have stated that you will provide repository information for your data at acceptance. Should your manuscript be accepted for publication, we will hold it until you provide the relevant accession numbers or DOIs necessary to access your data. If you wish to make changes to your Data Availability statement, please describe these changes in your cover letter and we will update your Data Availability statement to reflect the information you provide

4. We note that Figure (2) in your submission contain copyrighted images. All PLOS content is published under the Creative Commons Attribution License (CC BY 4.0), which means that the manuscript, images, and Supporting Information files will be freely available online, and any third party is permitted to access, download, copy, distribute, and use these materials in any way, even commercially, with proper attribution. For more information, see our copyright guidelines: http://journals.plos.org/plosone/s/licenses-and-copyright.

1. You may seek permission from the original copyright holder of Figure (2) to publish the content specifically under the CC BY 4.0 license. 

2. If you are unable to obtain permission from the original copyright holder to publish these figures under the CC BY 4.0 license or if the copyright holder’s requirements are incompatible with the CC BY 4.0 license, please either i) remove the figure or ii) supply a replacement figure that complies with the CC BY 4.0 license. Please check copyright information on all replacement figures and update the figure caption with source information. If applicable, please specify in the figure caption text when a figure is similar but not identical to the original image and is therefore for illustrative purposes only

Reviewers' comments:

Reviewer's Responses to Questions

**Comments to the Author**

1. Is the manuscript technically sound, and do the data support the conclusions?

Reviewer #1: Yes

Reviewer #2: Yes

2. Has the statistical analysis been performed appropriately and rigorously? 

Reviewer #1: Yes

Reviewer #2: Yes

3. Have the authors made all data underlying the findings in their manuscript fully available?

Reviewer #1: Yes

Reviewer #2: Yes

4. Is the manuscript presented in an intelligible fashion and written in standard English?

Reviewer #1: Yes

Reviewer #2: Yes

5. Review Comments to the Author

Reviewer #1: PONE-D-21-28218

Hauser, Frost, & Burns; “Ringed seal breeding habitat. ..”

Boveng review comments

This is a highly useful report on historical baseline measures of habitat for ringed seals, an abundant and important species that is vulnerable to loss of snow-covered sea-ice habitat in a rapidly warming Arctic. Although the study analyzes and presents for the first time a data set that was collected almost 40 years ago, it is very timely for ongoing efforts to understand the degree of risk posed to ringed seals by Arctic warming.

The work done in 1983-1984 to collect the data was undoubtably arduous; in situ studies of ringed seal habitat have been, and are likely to remain infrequent, so it is fortunate that the authors undertook the effort to ‘rescue’ the data and bring them into a form for contemporary analysis and sharing. The GIS and statistical methods appear to be appropriate and practical for the nature of the data.

The results add a reference baseline for quantifying ringed seal habitat characteristics in the eastern Chukchi Sea prior to discernible changes in sea ice and snow regimes, providing geographic contrast and comparison to studies done elsewhere in Arctic ringed seals’ range. The results corroborate some simplistic measures of habitat suitability that have been put forward previously, though the authors correctly note that the utility of these measures will remain somewhat tentative and difficult to gauge. Still, this will be a very helpful addition to the literature on the topic.

I found the interpretations of the results, the comparisons to related studies, and the assessment of this paper’s relevance to current ringed seal conservation topics to be appropriate and well-reasoned. I made minor comments and suggestions for the authors’ consideration, in markup on the attached copy of the manuscript. Most are likely to be of little consequence, though I do encourage consideration and perhaps discussion of two potential biases: One, marked as a comment at lines 228-231, is the possibility that late-season measurement of access hole size at grid 83-2 led to upward bias from melting at open structures. The second, marked at lines 381-382 is that differential (i.e., lower) detectability of breathing holes relative to lairs in areas of deformed ice could cause or contribute to biased proportions of structure types.

Congratulations on a nice paper.

Reviewer #2: This study offers important insight into the breeding habitat of ringed seals in the Chukchi Sea. The authors have made use of an historical dataset to describe breeding habitat and describe habitat dimensions and environmental conditions, specifically, ice and snow conditions at pup lairs and other structures, during a period prior to recent large-scale climatic changes in the Pacific Arctic. They have also described the spatial and temporal variability of structure densities. These summaries provide context for current conditions.

This study will help to contribute to our understanding of breeding habitat and serve as a baseline of information prior the environmental changes resulting from climate change. These findings may also be valuable for managers, specifically those involved with determining the status of ringed seals.

It is important that this paper is published. Overall, I have no substantive concerns, but I do have some suggestions, comments, and questions that I think will improve the manuscript.

General concerns throughout the manuscript:

Throughout, change use of “haulout lair(s)” to “haul-out lair(s)”. Used this way, “haul-out” is an adjective and should be hyphenated. Other examples include haul-out data, haul-out behavior. If it were used as a noun it would be one word, “haulout”. If it were used as a verb it would be two words, “haul out”. I’ve noted a few of the instances where this needs to be changed early in the manuscript in the line-by-line comments below but I did not identify every instance. I suggest searching the entire manuscript, including figure and table captions.

Punctuation used to cite references is not consistent; sometimes they are cited using parentheses “()” and others brackets “[]”. I’m sure the copy editor would identify these instances but the authors can correct these for the next submission. Please use the appropriate punctuation for this PLOS One and be consistent through-out the paper. For example, in the paragraph starting on line 460, through 477, brackets are used 4 times and parentheses used twice. Similar to my comment above, I’ve noted a few of the instances where this needs to be changed in the line-by-line comments below but I did not identify every instance.

The order of the methods as outlined by the subsection titles and headings is a logical progression. However, some paragraphs that describe the methods should be moved to other subsections better match the progression of steps and titles of the subsections should be changed to represent the new order. See comments below.

Please note my comments regarding the proper use of the terms “digital”, “digitized”, and “digitizing” versus “plotting” and “mapping” in reference to data used in a GIS as they have very specific meanings.

Some sentences are overly verbose and could be more succinct, particularly in the Discussion. Generally, the Discussion is long and the authors should look for opportunities to make this section more succinct. However, the authors did well to compare their results to studies of ringed seals in other areas. They also framed their findings in context to observed and predicted changes in environmental conditions. Although this is appropriate and an important aspect of this manuscript, the authors should scale back their pessimism and strong language regarding predictions. While some studies have identified changes in ringed seal health associated with climate change, these changes have not been universally negative. I do think describing these predictions is appropriate. However, describing such predictions with such certainty is outside the scope of this study.

Please feel free to contact me if you have any question.

Justin Crawford

Please see my review attached to this submission.

6. PLOS authors have the option to publish the peer review history of their article (what does this mean?). If published, this will include your full peer review and any attached files.

Reviewer #1: **Yes: **Peter Boveng

Reviewer #2: No

---

## [Author Response · Author response to Decision Letter 0]

11 Nov 2021

Reviewer 1: Peter Boveng

This is a highly useful report on historical baseline measures of habitat for ringed seals, an abundant and important species that is vulnerable to loss of snow-covered sea-ice habitat in a rapidly warming Arctic. Although the study analyzes and presents for the first time a data set that was collected almost 40 years ago, it is very timely for ongoing efforts to understand the degree of risk posed to ringed seals by Arctic warming.

The work done in 1983-1984 to collect the data was undoubtably arduous; in situ studies of ringed seal habitat have been, and are likely to remain infrequent, so it is fortunate that the authors undertook the effort to ‘rescue’ the data and bring them into a form for contemporary analysis and sharing. The GIS and statistical methods appear to be appropriate and practical for the nature of the data.

The results add a reference baseline for quantifying ringed seal habitat characteristics in the eastern Chukchi Sea prior to discernible changes in sea ice and snow regimes, providing geographic contrast and comparison to studies done elsewhere in Arctic ringed seals’ range. The results corroborate some simplistic measures of habitat suitability that have been put forward previously, though the authors correctly note that the utility of these measures will remain somewhat tentative and difficult to gauge. Still, this will be a very helpful addition to the literature on the topic.

I found the interpretations of the results, the comparisons to related studies, and the assessment of this paper’s relevance to current ringed seal conservation topics to be appropriate and well-reasoned. I made minor comments and suggestions for the authors’ consideration, in markup on the attached copy of the manuscript. Most are likely to be of little consequence, though I do encourage consideration and perhaps discussion of two potential biases: One, marked as a comment at lines 228-231, is the possibility that late-season measurement of access hole size at grid 83-2 led to upward bias from melting at open structures. The second, marked at lines 381-382 is that differential (i.e., lower) detectability of breathing holes relative to lairs in areas of deformed ice could cause or contribute to biased proportions of structure types.

Congratulations on a nice paper.

Thank you for the positive review of our motivation, approach, and interpretation. In our revisions, we have particularly considered Dr. Boveng’s comments about the possibility of upward bias on access hole measurements and detectability of breathing holes compared to lairs. Additional details are included below in response to Dr. Boveng’s direct comments on the draft manuscript.

Comments on the manuscript provided as a supplemental PDF:

Numerous grammatical edits that were suggested as comments on the manuscript have been completed. 

We additionally appreciate comments that included the need for additional response: 

• We added Frost et al. (2004) at line 79.

• We have added additional text to the Methods to describe how ice conditions were measured in the field.

• We have added more information on how we assess parametric assumptions and applied transformations in the statistical analysis methods section.

• In the statistical analysis methods section, Dr. Boveng notes: “Were these lairs included in the measurements of hole diameters? If so, it seems like they could be biased upward because, in my limited experience, the margins of holes seem to melt and enlarge rapidly once the lair opens up. I note that breathing holes in Kotzebue Sound H and P lairs were considerably larger than those in Ledyard Bay, and I wonder if the conditions at 83-2 could be the explanation.” 

Yes, this is a good point. In further consideration, we do find that breathing holes in Grid 83-2 are significantly larger than those in Grid 83-1 (ANOVA, p= 0.005, see figures attached in 'Response to Reviewers'). As Dr. Boveng points out, this is likely due to the advanced stage of melt. Grid 83-2 was also surveyed later in the year than most other grids. Later in the season, the structures have received more use by female and pup and it is likely there are larger due to use. This includes access holes. We now report updated findings on Grid 83-2 breathing hole diameters in the Results and have added more details to the Discussion to clarify and discuss differences between grids and associated impacts on interpretation.

• We have consistently revised to ‘haul-out’ lairs throughout.

• Thanks for pointing out the new Kovacs et al. 2021 paper comparing ringed seal sizes. We have added it to our Discussion.

• In the Discussion, Dr. Boveng asks: “Do you think it is possible that breathing holes are less detectable by dogs, eg, maybe they have less scent because the seals' whole bodies are not in contact with the cavity surfaces, etc? If so, the differential detectability might be even stronger in the areas of deformed ice that are optimal for lairs; a breathing hole under a jumble of ice might tend to go undetected. Then the apparent difference in typical surroundings for holes versus lairs might not be real.

We have added text to note this potential limitation to our study approach, but also describe that previous work indicated no bias in the types of structures located by each of the dogs that were also used in this study (Frost and Burns 1989). Our work and other studies suggest that breathing holes may actually be easier to detect because they are often along linear features and flat ice with less snow cover. 

• In the Discussion, Dr. Boveng raises some ideas about why pup and haul-out lairs may have larger access holes, but notes that this is just speculative. We are unaware of any evidence that mothers “grab” their pups to escape predators and did not collect information to support or refute this suggestion, so we feel that this is too speculative to include in the Discussion. 

Reviewer #2: Justin Crawford

This study offers important insight into the breeding habitat of ringed seals in the Chukchi Sea. The authors have made use of an historical dataset to describe breeding habitat and describe habitat dimensions and environmental conditions, specifically, ice and snow conditions at pup lairs and other structures, during a period prior to recent large-scale climatic changes in the Pacific Arctic. They have also described the spatial and temporal variability of structure densities. These summaries provide context for current conditions.

This study will help to contribute to our understanding of breeding habitat and serve as a baseline of information prior the environmental changes resulting from climate change. These findings may also be valuable for managers, specifically those involved with determining the status of ringed seals.

It is important that this paper is published. Overall, I have no substantive concerns, but I do have some suggestions, comments, and questions that I think will improve the manuscript.

Thank you for the positive review of our study as well as your suggested improvements to the manuscript. We have considered your edits as noted in more detail below.

General concerns throughout the manuscript:

Throughout, change use of “haulout lair(s)” to “haul-out lair(s)”. Used this way, “haul-out” is an adjective and should be hyphenated. Other examples include haul-out data, haul-out behavior. If it were used as a noun it would be one word, “haulout”. If it were used as a verb it would be two words, “haul out”. I’ve noted a few of the instances where this needs to be changed early in the manuscript in the line-by-line comments below but I did not identify every instance. I suggest searching the entire manuscript, including figure and table captions.

We have carefully revised throughout. 

Punctuation used to cite references is not consistent; sometimes they are cited using parentheses “()” and others brackets “[]”. I’m sure the copy editor would identify these instances but the authors can correct these for the next submission. Please use the appropriate punctuation for this PLOS One and be consistent through-out the paper. For example, in the paragraph starting on line 460, through 477, brackets are used 4 times and parentheses used twice. Similar to my comment above, I’ve noted a few of the instances where this needs to be changed in the line-by-line comments below but I did not identify every instance.

We updated our EndNote template for PLOS One to adjust the citation format that uses brackets. 

The order of the methods as outlined by the subsection titles and headings is a logical progression. However, some paragraphs that describe the methods should be moved to other subsections better match the progression of steps and titles of the subsections should be changed to represent the new order. See comments below.

We have reorganized and changed titles on the subsections within the Methods based on your suggestions.

Please note my comments regarding the proper use of the terms “digital”, “digitized”, and “digitizing” versus “plotting” and “mapping” in reference to data used in a GIS as they have very specific meanings.

We have revised following your suggestions, and we no longer include “digital,” “digitized,” or “digitizing.”

Some sentences are overly verbose and could be more succinct, particularly in the Discussion. Generally, the Discussion is long and the authors should look for opportunities to make this section more succinct. However, the authors did well to compare their results to studies of ringed seals in other areas. They also framed their findings in context to observed and predicted changes in environmental conditions. Although this is appropriate and an important aspect of this manuscript, the authors should scale back their pessimism and strong language regarding predictions. While some studies have identified changes in ringed seal health associated with climate change, these changes have not been universally negative. I do think describing these predictions is appropriate. However, describing such predictions with such certainty is outside the scope of this study.

We have revised following your suggestions to simplify, improve clarity, delete extraneous words and sentences, and tone down language regarding predictions.

Please feel free to contact me if you have any question.

Justin Crawford

Please see my review attached to this submission.

Thanks for the many editorial suggestions. We have incorporated your line comments and edits. We note responses to the few cases where we chose not to incorporate your suggestions below.

• We did not remove “Materials” from the heading “materials and methods”, because this is the section heading format for the journal.

• We did not move the paragraph describing snow accumulation on ice from the Methods ‘Study Area’ section to the Introduction, because we think this is important additional detail to the ice and snow conditions in our study area and during the timing of our study. However, we did add additional context to the Introduction, at the suggested paragraph, about snow accumulation on ice to describe how those conditions affect lair construction.

• We did not include summary statistics of the snow and ice measurements for combined study regions in the Results. As noted in the Methods, we were concerned about mixing environmental conditions sampled in different years and regions of the Chukchi Sea. However, we do provide summary statistics of lair measurements for study regions combined because the dimensions of a lair corresponds to the size of seals and not variable environmental conditions.

• We did not include lines on Fig 3 to summarize overall measurement comparisons, because the figures will become too complicated. As noted by Mr. Crawford, comparisons are described in the text and summary values are reported in Tables 3 and 4.

---

## [Editor Report · Decision Letter 1]

15 Nov 2021

Ringed seal (Pusa hispida) breeding habitat on the landfast ice in northwest Alaska during spring 1983 and 1984

PONE-D-21-28218R1

Dear Dr. Hauser,

We’re pleased to inform you that your manuscript has been judged scientifically suitable for publication and will be formally accepted for publication once it meets all outstanding technical requirements.

Kind regards,

William David Halliday, Ph.D.

Academic Editor

PLOS ONE
---

## [Editor Report · Acceptance letter]

18 Nov 2021

PONE-D-21-28218R1 

Ringed seal (*Pusa hispida*) breeding habitat on the landfast ice in northwest Alaska during spring 1983 and 1984 

Dear Dr. Hauser:

I'm pleased to inform you that your manuscript has been deemed suitable for publication in PLOS ONE. Congratulations! Your manuscript is now with our production department. 

Kind regards, 

on behalf of

Dr. William David Halliday 

Academic Editor

PLOS ONE